# Folic Acid and Vitamin B12 Prevent Deleterious Effects of Rotenone on Object Novelty Recognition Memory and *Kynu* Expression in an Animal Model of Parkinson’s Disease

**DOI:** 10.3390/genes13122397

**Published:** 2022-12-17

**Authors:** Gabriela Canalli Kretzschmar, Adriano D. S. Targa, Sheila Coelho Soares-Lima, Priscila Ianzen dos Santos, Lais S. Rodrigues, Daniel A. Macedo, Luis Felipe Ribeiro Pinto, Marcelo M. S. Lima, Angelica Beate Winter Boldt

**Affiliations:** 1Laboratory of Human Molecular Genetics, Department of Genetics, Federal University of Paraná (UFPR), Centro Politécnico, Jardim das Américas, Curitiba 81531-990, PR, Brazil; 2Postgraduate Program in Genetics, Department of Genetics, Federal University of Paraná (UFPR), Centro Politécnico, Jardim das Américas, Curitiba 81531-990, PR, Brazil; 3Laboratory of Neurophysiology, Department of Physiology, Federal University of Paraná (UFPR), Centro Politécnico, Jardim das Américas, Curitiba 81531-990, PR, Brazil; 4Molecular Carcinogenesis Program, National Cancer Institute, Research Coordination, Rio de Janeiro 20231-050, RJ, Brazil

**Keywords:** tryptophan pathway, DNA methylation, gene expression, neurodegenerative diseases, *Kynu*, cognitive impairment

## Abstract

Parkinson’s disease (PD) is characterized by a range of motor signs, but cognitive dysfunction is also observed. Supplementation with folic acid and vitamin B12 is expected to prevent cognitive impairment. To test this in PD, we promoted a lesion within the substantia nigra *pars compacta* of rats using the neurotoxin rotenone. In the sequence, the animals were supplemented with folic acid and vitamin B12 for 14 consecutive days and subjected to the object recognition test. We observed an impairment in object recognition memory after rotenone administration, which was prevented by supplementation (*p* < 0.01). Supplementation may adjust gene expression through efficient DNA methylation. To verify this, we measured the expression and methylation of the kynureninase gene (*Kynu*), whose product metabolizes neurotoxic metabolites often accumulated in PD as kynurenine. Supplementation prevented the decrease in *Kynu* expression induced by rotenone in the substantia nigra (*p* < 0.05), corroborating the behavioral data. No differences were observed concerning the methylation analysis of two CpG sites in the *Kynu* promoter. Instead, we suggest that folic acid and vitamin B12 increased global DNA methylation, reduced the expression of *Kynu* inhibitors, maintained Kynu-dependent pathway homeostasis, and prevented the memory impairment induced by rotenone. Our study raises the possibility of adjuvant therapy for PD with folic acid and vitamin B12.

## 1. Introduction

Parkinson’s disease (PD) is the second most common chronic neurodegenerative disease [1]. It is characterized by intraneural α-synuclein accumulation, which contributes to the loss of dopaminergic neurons in the substantia nigra *pars compacta* (SNpc) [2]. This ultimately decreases the dopamine release in the dorsal striatum tissue (St), leading to well-known motor deficits. Several non-motor signs precede the motor symptoms, e.g., cognitive deficits, olfactory dysfunction, sleep abnormalities, cardiac sympathetic denervation, constipation, depression, and pain [1,3,4,5,6,7,8,9].

The disease may occur due to the monogenic mendelian inheritance of some genetic variants (early-onset familial form). Nonetheless, most cases are sporadic and result from a complex interplay between genetic, epigenetic, and environmental factors. Among the epigenetic factors, aging has a prominent role in dysregulating gene methylation patterns, particularly in the brain [10]. Gene methylation occurs through a DNA methyltransferase (DNMT), which covalently transfers a methyl group from S-adenosyl-L-methionine (SAM) to the carbon 5 of cytosine within a 5′CpG3′ dinucleotide, converting it to a 5-methylcytosine (5mC) [11]. DNA methylation influences gene expression through different mechanisms. Those best described include interference with the recognition of DNA motifs by transcription factors (TFs) and the recruitment of specific methyl-CpG-binding proteins, which further recruit co-repressor protein complexes and mediate gene silencing on promoter regions [12,13]. Thus, the homeostasis of this process is fundamental for health [11]. Hypomethylation of the promoter region has already been found for the PD-associated gene *SNCA* (synuclein alpha), leading to a pathological increase in α-synuclein expression [14].

SAM production depends on adequate folate and complex B vitamin supply for the one-carbon cycle [12]. Deregulating this metabolic cycle results in high homocysteine levels, SAM shortage, and DNA hypomethylation. Elevated homocysteine levels have already been reported in PD patients and have a toxic effect on dopaminergic neurons in animal models, being associated with the declined function of peripheral nerves, balancing disturbances, and cumulating levodopa dosage [15,16,17,18]. Indeed, some studies with folic acid and vitamin B12 supplementation in PD patients and animal models have been proposed to improve cognition and cholinergic transmission, prevent dopaminergic degeneration, ameliorate mitochondrial dysfunction and oxidative stress, prevent low metabolic activity and locomotor defects, and even reduce blood levels of inflammatory cytokines, being more effective if administered together [19,20,21,22,23,24,25,26,27,28,29,30,31,32]. However, some reports failed to demonstrate an effect [33,34,35]. In this sense, there is an urgent need for studies with animal models to demonstrate the potential role of folates in cognitive improvement [36]. An animal model is defined as an experimental animal that has a disease or injury similar to the human condition by means of a pharmacological/neurotoxic induction [37]. In this sense, rotenone offers a well-established model that mimics an early stage of PD with different features, such as anxiety [38], depressive-like behavior [39], olfactory dysfunction [6,40], cognitive impairment [41], and sleep disturbances [9].

The kynurenine pathway (KP) is the main available tryptophan-processing route [42,43]. It regulates the immune response and a variety of neurotransmitter and cognitive functions, being associated with cancer and autoimmune, inflammatory, psychiatric, and neurodegenerative diseases [44,45,46,47,48,49,50,51,52,53,54,55,56].

After KP’s first association with PD in 1992 by Ogawa and colleagues [57], several authors reported the involvement of KP-associated enzymes and metabolites with the disease, being a possible biomarker and therapeutic target for PD [58,59,60,61,62,63,64,65,66]. In a previous study [67], we observed that rotenone administration in rats increased their plasmatic kynurenine (KYN) levels. Moreover, it was already demonstrated that the tryptophan levels in the PD animal models decreased while the kynurenine levels increased, concluding that KP is involved in PD [63]. Within the KP, KYN results from the tryptophan metabolism [52], and KYNU is one of the enzymes responsible for KYN and 3-hydroxykynurenine (3-HK) processing. It is encoded by the kynureninase (*KYNU*) gene and has already been reported to play a pro-inflammatory role in human diseases [58,68]. Therefore, we hypothesized that a change in *Kynu* gene expression and improved memory consolidation would result from dietary supplementation with folic acid and vitamin B12 since these nutritional elements are critical for the epigenetic regulation of gene expression. The results of our study may support the therapeutic benefit arising from this adjuvant nutritional supplementation in PD treatment.

## 2. Materials and Methods

### 2.1. Animals

The study was approved by the ethics committee of the UFPR—Federal University of Paraná (approval ID #1289) and carried out following the Guidelines of Ethics and Experimental Care and Use of Laboratory Animals of National Institutes of Health (NIH Publications No. 8023, revised 1978) (SBCAL).

This study was performed with 33 male Wistar rats (*Rattus norvegicus*), three months old, weighing approximately 280–320 g, from the UFPR bioterium. The animals were maintained in groups of five individuals within polypropylene cages in a temperature-controlled room (22 ± 2 °C) with a 12:12 h light:dark cycle (lights on 7:00 AM). Bottles of water and food pellets were freely available throughout the experiment.

We made all efforts to minimize animal suffering according to the recommendations of the UFPR ethics committee. After the stereotaxic surgery, the animals were maintained in an appropriate room to recover, with constant monitoring to observe whether they could drink and eat. During the following days, the weight of the animals was continuously monitored. All efforts were made to reduce the number of animals as well. According to previous reports, we kept this number to the minimum necessary to not compromise the statistical analysis [41].

### 2.2. Experimental Design

The study involved exploratory research. On day 0, the animals were submitted to stereotaxic surgery for rotenone or dimethylsulfoxide (DMSO) administration within the SNpc. The supplementation with folic acid and vitamin B12 or saline solution started the following day and lasted 14 days. Thus, through simple randomization, the 33 animals were distributed into three groups: two without supplementation (WS)—rotenone (R_WS) and sham (S); one with supplementation (Sup)—rotenone (R_Sup). The object recognition task test (ORT) habituation phase took place on days 10, 12, and 14. After the last habituation (day 14), the animals performed the training of ORT, and, on the following day (day 15), they completed the test phase of ORT (N = 33). Euthanasia was performed after the test (day 15) using a guillotine. Samples from St and substantia nigra (SN) were collected and maintained at −80 °C until processing (Figure 1A).

### 2.3. Stereotaxic Surgery

The animals were sedated with intraperitoneal xylazine (10 mg/Kg; Syntec do Brasil Ltda.,Cotia, SP, Brazil) and anesthetized with intraperitoneal ketamine (90 mg/Kg; Syntec do Brasil Ltda., Cotia, SP, Brazil) 15 min before surgery (8:00–11:00 AM). The choice of anesthetic was based on previous studies with this model [9,41]. For rotenone infusion within the SNpc, we used bregma as a reference for the following coordinates: (AP) = −5.0 mm, (ML) = +2.1, (ML) = −2.1 mm, and (DV) = −8.0 mm [69]. Rotenone (12 mg/mL; Sigma-Aldrich, St. Louis, MO, USA) or DMSO 10% *v*/*v* (Sigma-Aldrich, St. Louis, MO, USA) bilateral infusions were performed using an electronic infusion pump (Insight Instruments, Ribeirão Preto, Brazil) at a rate of 0.33 mL/min for 3 min [8].

### 2.4. Supplementation Procedure

The vitamins used for supplementation were obtained from a compounding pharmacy. The powdered pharmaceutical forms were diluted in saline solution (0.9%) for administration through an 8 cm orogastric needle. The animals were supplemented by gavage with either folic acid/vitamin B12 (5 and 0.5 mg/Kg, respectively) or saline (0.9%), performed once a day (7:30 AM–10:30 AM) for 14 days after the surgery. Such dosages were adopted to ensure the occurrence of a significant central nervous system effect [70] associated with the occurrence of purported effects on gene expression [71], particularly in the *Kynu* gene.

### 2.5. Object Recognition Task Test (ORT)

We used the ORT to investigate the memory consolidation process [72]. The apparatus consisted of an open box (width × length × height = 60 cm × 60 cm × 50 cm) made of wood and covered with a black opaque plastic film. The objects to be discriminated against were made of biologically neutral materials such as glass, plastic, or metal. The procedure consisted of three phases: habituation, training, and the test (performed from 8:00 to 10:00 AM). The animals had three minutes in the habituation phase on days 10, 12, and 14 to explore the arena without the objects. During the training phase (15 min after habituation on day 14), two identical objects were exposed in the back corners of the open box, 10 cm away from the sidewall. The rat was placed in the open box facing away from the objects, and after 3 min of exploration, the animal was removed from the open box and returned to its cage. Twenty-four hours later (test phase, 3 min of duration), two objects were presented in the same locations occupied by the previous sample objects. One object was identical to the object seen in the training phase (familiar object), and the other was different (new object). Rats are animals of exploratory behavior, so it is expected that an animal with intact memory remembers the familiar object, spending more time exploring the new one. The tests were video-recorded and analyzed by an experimenter blind to the treatments. It was considered exploration only when the rat touched the object with its nose or when the rat’s nose was directed toward an object at a distance ≤2 cm. Based on previous laboratory studies, we assumed a cut-off point of at least 10 s of total object exploration [73].

### 2.6. Gene Regulation and Expression Analysis

#### 2.6.1. DNA Methylation

In total, 33 DNA samples (Figure 1B) were extracted from the frozen brain tissue of SN and St, with the Wizard Genomic DNA Purification Kit (Promega, Madison, Wisconsin, EUA #A1125), following the manufacturer’s instructions. The DNA was treated with the *EpiTect* 96 bisulfite kit (Qiagen, Hilden, Germany #59110) to convert unmethylated cytosines into uracil. We measured the methylation of two evolutionarily conserved CpGs in the promoter region of the *Kynu* gene in the rat genome (RGSC 6.0/rn6). One was orthologous to *cg15836722* in the *KYNU* human promoter, reported by Roberson and colleagues [74] as differentially methylated in skin samples of psoriasis patients. The primers were designed using PyroMark Assay Design 2.0 (Qiagen). The primer sequences and their positions in the flanking *Kynu* sequence are listed in Appendix A.

PCR was performed using a biotinylated reverse primer (Qiagen custom assay, Hilden, Germany) and the PyroMark PCR Kit (Qiagen, Hilden, Germany #978703) (protocol in Appendix A), followed by 1.5% agarose (Uniscience, Osasco, SP, Brazil #UNI-R10113) gel electrophoresis to check the amplicon’s quality and length. Pyrosequencing was performed in the Qiagen PyroMark Q96 ID System (RRID: SCR_020413), following the manufacturer’s instructions. The percentage of DNA methylation per site was calculated using the PyroMark Q96 ID Software 2.5 (Qiagen, Hilden, Germany).

#### 2.6.2. Gene Expression

Total RNA was extracted from 21 samples (Figure 1B) from frozen brain tissue (SN and St), with the ReliaPrep miRNA Cell and Tissue Miniprep System Kit (Promega, Madison, Wisconsin, EUA #Z6212), following the manufacturer’s instructions. The RNA was reverse-transcribed into cDNA with the High-Capacity cDNA Reverse Transcription Kit (Applied Biosystems, Waltham, Massachusetts, EUA #4368813). The *Kynu* primers were designed to be complementary to exon 13–14 junctions (Appendix A). The relative mRNA levels were normalized using the median of mRNA expression of two endogenous genes, beta-actin (*Actb*) and hypoxanthine phosphoribosyltransferase 1 (*Hprt1*), using the sequences of the primers already described by Elfving and colleagues [75]. *Kynu* mRNA levels were measured in quantitative real time using the GoTaq qPCR Master Mix (Promega, Madison, Wisconsin, EUA #A6001) (protocol in Appendix A) in the ViiA 7 Real-Time PCR System (Applied Biosystems, Waltham, Massachusetts, EUA #4453536). All assays were conducted in triplicate, and Cq values (threshold cycle) were calculated using the QuantStudio Real-Time PCR Software version 1.3 (Applied Biosystems, Waltham, Massachusetts, EUA). For calculating fold-change values of gene expression, we used the comparative Cq method 2^−ΔΔCq^ [76]. The *Kynu* fold change value was compared using the Mann–Whitney and Kruskal–Wallis tests between the treatment groups and sham for each brain region separately.

### 2.7. Statistical Analysis

Statistical analyses were performed using GraphPad Prism v.6 software (GraphPad Prism v.6 Software, La Jolla, CA, USA). The data were tested for normality using the D’Agostino and Pearson test. Parametric data were expressed as mean (SD) and non-parametric as median with interquartile range. For analysis of the methylation degree, gene expression, and memory consolidation, the groups were compared using the appropriate test for parametric and non-parametric data (unpaired *t*-test and two-way ANOVA for parametric data; Mann–Whitney and Kruskal–Wallis tests for non-parametric data). Outliers were defined for ORT as animals with less than 10 s of object exploration. We did not record outliers for other analyses. For correlation analyses, we used Spearman’s test. All *p*-values were corrected for multiple testing using the false discovery rate (FDR) method [77], performed in R language 3.6.1, through the Stats package [78], or with Sidak’s multiple comparisons tests. Corrected *p*-values lower than 0.05 were considered significant. We used the UCSC Genome Browser [79], TargetScan Browser [80], TarBase v.8 [81], miRGate [82], LNCipedia (v.5.2) [83], and NONCODE [84] to search for possible regulatory factors of the *Kynu* gene.

## 3. Results

### 3.1. Object Recognition Task Test (ORT)

The animals in the sham group spent more time exploring the new object compared to the familiar one (*p* < 0.05). However, we did not observe a statistical difference concerning the exploration of objects in the R_WS group, suggesting a memory impairment caused by rotenone administration. Interestingly, the supplementation with folic acid and vitamin B12 prevented this deleterious effect *(p* < 0.001) (Figure 2).

### 3.2. Kynu Expression

*Kynu* expression was lower in the SN of R_WS rats, compared to both sham (*p* < 0.05, fold change = −2.45) and R_Sup groups (*p* < 0.05, fold change = −2.50) (Figure 3A). *Kynu*’s gene expression did not differ between the supplemented and sham animals, suggesting that (1) rotenone administration suppressed *Kynu’s* expression and (2) supplementation prevented the rotenone-induced decrease in *Kynu*’s gene expression. The supplementation was not associated with differences in *Kynu* expression within the St (Figure 3B).

Methylation levels differed between the *Kynu* CpGs in the sham group, both in the SN and between the investigated brain regions (*p* < 0.01 in both cases). However, the methylation patterns of these were not correlated with the levels of *Kynu* expression or with the supplementation (Appendix A).

## 4. Discussion

In this study, we observed that rotenone administration impaired the consolidation of object recognition memory and decreased the expression of *Kynu*. Interestingly, supplementation with folic acid and vitamin B12 prevented the rotenone-induced memory impairment and the downregulation of *Kynu* expression. We did not observe the effects of rotenone or supplementation with folic acid and vitamin B12 concerning the methylation levels in the two *Kynu* CpG sites analyzed.

Disturbances in the one-carbon cycle, leading to changes in SAM production and, consequently, in methylation patterns, have already been associated with PD and linked to a cognitive decline [16]. PD patients with cognitive deficits have high levels of homocysteine (hyperhomocysteinemia) and reduced levels of folate and vitamin B12 in plasma, serum, and cerebrospinal fluid [18,22,32,85,86,87,88,89], especially those treated with levodopa (L-DOPA), which leads to side effects such as dyskinesia over the years [90]. The metabolism of plasma levodopa involves methylation reactions catalyzed by catechol-O-methyltransferase (COMT), which uses SAM as a methyl donor, resulting in S-adenosylhomocysteine (SAH) and consequent homocysteine formation [22,91,92]. The accumulation of homocysteine is toxic to dopaminergic neurons, contributing to neurodegeneration [17,93], which can be avoided by promoting SAH recycling within the one-carbon cycle. Considering this, vitamin B12 and folate are essential, given that they act as cofactors for the remethylation of homocysteine in methionine, which can be used for SAM production [22]. Thus, a possible strategy to counteract homocysteine accumulation is by supplementing the diet with essential elements for the optimal functioning of this cycle, such as vitamin B12 and folic acid [28,30,94,95,96,97].

We observed that rotenone administration impaired memory consolidation in the ORT, as previously demonstrated [41,67]. Growing evidence shows an increase in the importance of cognitive impairment in PD. This impairment is also due to increasing life expectancy and directly impacts the quality of life of affected individuals [98,99]. Moreover, cognitive decline may be considered a relevant marker of disease evolution and, together with other non-motor disturbances, renders PD much more complex than merely a motor disease [100]. Additionally, the rotenone-induced model of PD is typically an early-phase model, which means that its most important feature is to produce a mild set of nigrostriatal lesions. As a result, we have a model that is excellent for recapitulating the nonmotor disturbances and memory dysfunctions of PD. Interestingly, folic acid and vitamin B12 supplementation prevented the decrease in *Kynu* expression induced by this neurotoxin and hindered its deleterious effect on recognition memory. Our results corroborate a study that analyzed the impact of supplementation with folic acid and vitamin B12 on memory and the levels of inflammatory factors in 240 patients with mild cognitive impairment (MCI) [19]. They found an overall improvement in cognitive function, and reduced levels of homocysteine, interleukin 6 (IL-6), tumor necrosis factor-alpha (TNF-α), and monocyte chemotactic protein 1 (MCP-1). The KYN pathway can be activated by inflammatory factors, such as TNF-α and IL-6, which lead to the expression of indoleamine 2,3-dioxygenase 1 (IDO-1), inducing the catalysis of tryptophan through this route [42,43,54]. Moreover, inflammatory mediators can also stimulate other enzymes involved in the pathway, such as kynurenine 3-monooxygenase (KMO), which leads to a high concentration of neurotoxic metabolites [101]. A reduction in these inflammatory factors by supplementation could reduce the production of neurotoxic metabolites, leading to memory improvement.

In our study, rotenone administration, folic acid, and vitamin B12 supplementation were associated with differences in *Kynu* expression. Within the SN, the expression of this gene decreases as a direct consequence of rotenone injection. The downregulating pathway is interrupted by vitamin B12 and folic acid supplementation, most probably due to the higher provision of SAM in the one-carbon cycle. The supplementation maintained *Kynu*’s expression at basal levels, similar to those of the sham group. Interestingly, *Kynu*’s expression in the St presented no changes, which may be related to the fact that this region is the terminal portion of the nigrostriatal pathway, where the regulatory impact of treatment probably manifests later [102].

Despite the changes in gene expression, the methylation levels of the analyzed CpG sites in the *Kynu* promoter remained unaltered. *Kynu* is a low CpG promoter gene, with only sparse CpG dinucleotides, indicating that the regulation of this gene may involve other mechanisms besides methylation, such as regulatory proteins [103]. *Kynu*’s promoter harbors binding motifs for different transcription repressor proteins—CCAAT enhancer-binding protein beta (CEBPB), FOS, GATA3, JUN, JUND, MYC, and RUNX3 (UCSC). These proteins are encoded by genes with extensive CpG islands in their promoter regions and/or along the gene. Thus, in contrast to *Kynu*, these genes are more likely to be regulated by their CpG methylation levels. Although supplementation did not alter *Kynu*’s methylation at two representative CpG sites, it possibly changed the methylation of repressor protein genes, silencing their expression and consequently liberating *Kynu*’s promoter region for transcription. A similar scenario may be true for the expression of repressor non-coding RNAs (ncRNAs), as microRNAs (miRNA), controlling the translation of *Kynu*’s mRNA in the cytoplasm. As an example, miR-142 downregulation has already been correlated with *KYNU*’s upregulation in humans [104]. Nevertheless, the genes encoding miRNAs that potentially regulate Kynu expression in rats do not present CpG islands (UCSC), and the miR-142 rat homolog does not regulate the *Kynu* gene (TargetScan). At the same time, the information available in the literature and the databases of long non-coding RNAs (lncRNAs) in rats are scarce. There is still no evidence that lncRNAs might regulate *Kynu* in any of the consulted databases.

Based on this, we propose that, in a non-pathological scenario, *Kynu* is possibly expressed at baseline levels, as well as the putative repressor protein/ncRNA, which is not expressed in sufficient levels to suppress *Kynu* expression completely (Figure 4A). However, in the specific context of the rotenone-induced PD model, this repressor may have its promoter region hypomethylated, leading to continuous expression and increasing the protein/ncRNA levels, resulting in intense *Kynu* repression (Figure 4B). With folic acid and vitamin B12 supplementation, the repressor’s promoter region returns to its baseline methylation state or becomes completely methylated, reducing or silencing the expression of this repressor and allowing *Kynu*’s expression (Figure 4C).

Fagotti et al. [67] recently reported the accumulation of KYN in plasma after similar nigrostriatal rotenone-induced lesions in rats. The kynurenine pathway in PD contributes to inflammation, suggesting that KYN might be an early-phase biomarker of PD [101]. KYN tends to be preferentially processed by KMO [101,105]. Moreover, 3-HK, a neuroactive metabolite resulting from this processing (Figure 5), has already been observed at high levels in the frontal cortex, putamen, SN [57], and cerebrospinal fluid [106] of PD patients. In addition, low levels of 3-hydroxy anthranilic acid, a product originating from 3-HK processing by KYNU, have been observed in PD, suggesting that KYNU activity is reduced in patients [58]. This process can result in the accumulation of 3-HK, free radicals, superoxide, and hydrogen peroxide, which cause oxidative damage, neuronal apoptosis, and neurodegeneration [50,101,107,108]. Oxidative stress is indeed known to play one of the leading roles in PD-induced neurodegeneration, as reviewed in [109]. Therefore, methods to avoid or reduce this process should be urgently evaluated.

This research has some limitations. We did not quantify 5-hydroxymethylcytosine, which is an intermediate in the demethylation process. Moreover, we considered only two CpG sites in our *Kynu* methylation analysis. We did not assess KYNU protein, 3-HK metabolite, and homocysteine levels in this study. We did not evaluate the levels of gene expression and methylation status of possible *Kynu* repressors. However, the influence of the KP in PD is increasingly evident. Deregulations due to lower expression of its essential enzymes can lead to the accumulation of neurotoxic metabolites and inflammation, causing neurodegeneration. Our promising results from a validated animal model of PD contribute to an increase in knowledge about the influence of folates at the molecular and behavioral levels, a field that urgently needs further research.

## 5. Conclusions

Considering the results obtained in our study, we suggest that supplementation with folic acid and vitamin B12 may contribute to counteracting the neurodegenerative process, and we encourage studies with other PD models and clinical trials with this approach.

## Figures and Tables

**Figure 1 genes-13-02397-f001:**
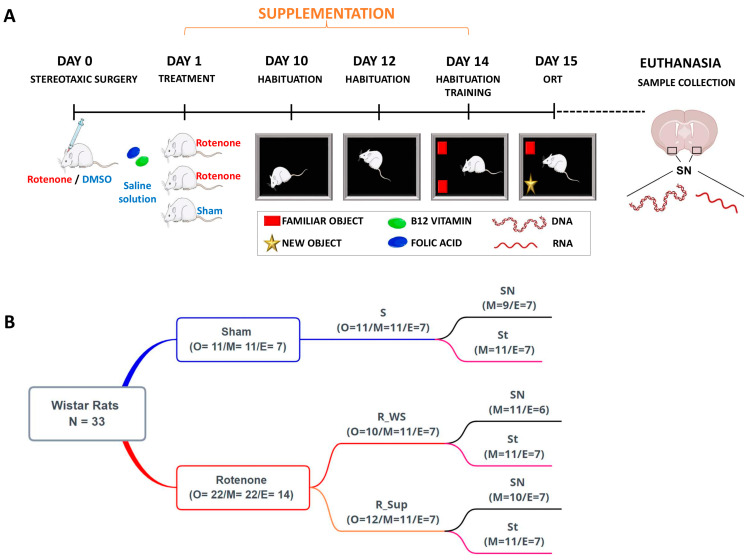
Experimental design. (**A**) The animals were subjected to stereotaxic surgery for rotenone or dimethylsulfoxide (DMSO) administration within the substantia nigra *pars compacta*. The rats were distributed into three groups: two without supplementation (WS)—rotenone (R_WS) and sham (S); one with B12 vitamin and folic acid supplementation (Sup)—rotenone (R_Sup). The animals were subjected to the object recognition task test (ORT). For molecular analysis, DNA and RNA were extracted from the substantia nigra (SN) and the striatum (St). (**B**) Animal numbers for each experiment. It is important to note that n = 33 corresponds to the maximum number of animals used, but only 21 samples were available in the expression analysis. Within parentheses: number of individuals for each experiment. O = object recognition task; M = methylation pattern analysis; E = gene expression analysis.

**Figure 2 genes-13-02397-f002:**
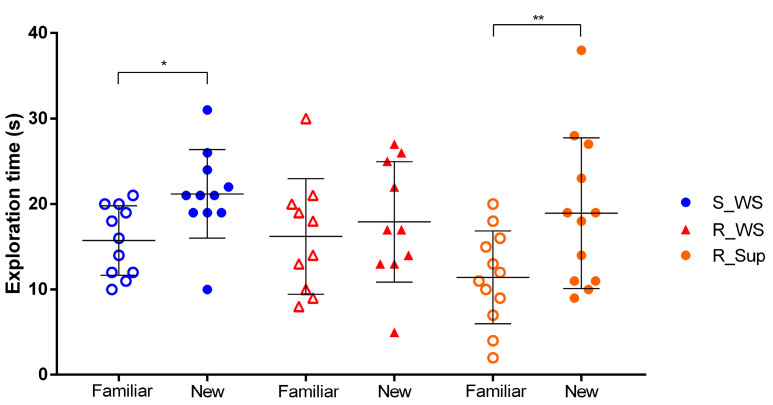
Exploration time of familiar and new objects in the object recognition task (ORT). The graphic shows the exploration time of each object—familiar or new—among the experimental groups. Animals from the sham group (S: F- ± 4.1, N- ± 5.2, n = 11) spent more time exploring the new object, as expected—* *p* < 0.05. Rotenone administration (R_WS: F- ± 6.8, N- ± 7.0, n = 10) led to an impairment in object recognition memory (*p* = 0.80), which was prevented in supplemented animals (R_Sup: F- ± 5.4, N- ± 8.8, n = 12)—** *p* < 0.01. Data are expressed as mean (SD).

**Figure 3 genes-13-02397-f003:**
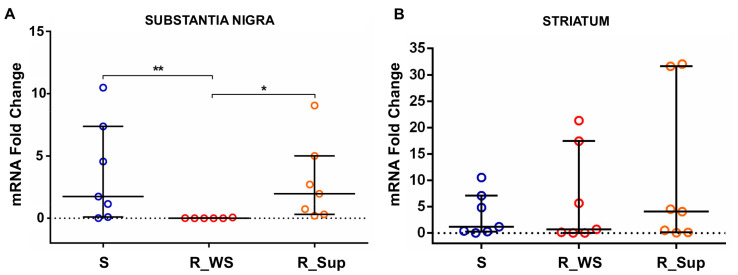
Differences in *Kynu* expression between the groups in the substantia nigra and striatum. (**A**) *Kynu* expression in rotenone-treated rats without supplementation (R_WS—median: 0.006, min: 0.002, max: 0.062, n = 6) is practically nonexistent when compared to the sham group (** *p <* 0.05, fold change = −2.45). Those rotenone-treated rats that received the supplementation with folic acid and vitamin B12 (R_Sup—median = 1.97, min: 0.202, max: 9.05, n = 7) maintained the expression levels observed for the sham group (S—median = 1.74, min: 0.014, max: 10.49, n = 7), meaning that the rotenone-induced decrease in *Kynu*’s gene expression was prevented in the supplemented group (R_Sup) (* *p* < 0.05, fold change = −2.50). (**B**) There were no significant changes in *Kynu* expression between groups in the striatum. R_Sup—median = 4.05, min: 0.06, max: 32.06, n = 7; R_WS—median = 0.69, min: 0.03, max: 21.33, n = 7; S—median = 1.19, min: 0.02, max: 10.52, n = 7. Data are expressed as median (IQR). Min = minimum; max = maximum.

**Figure 4 genes-13-02397-f004:**
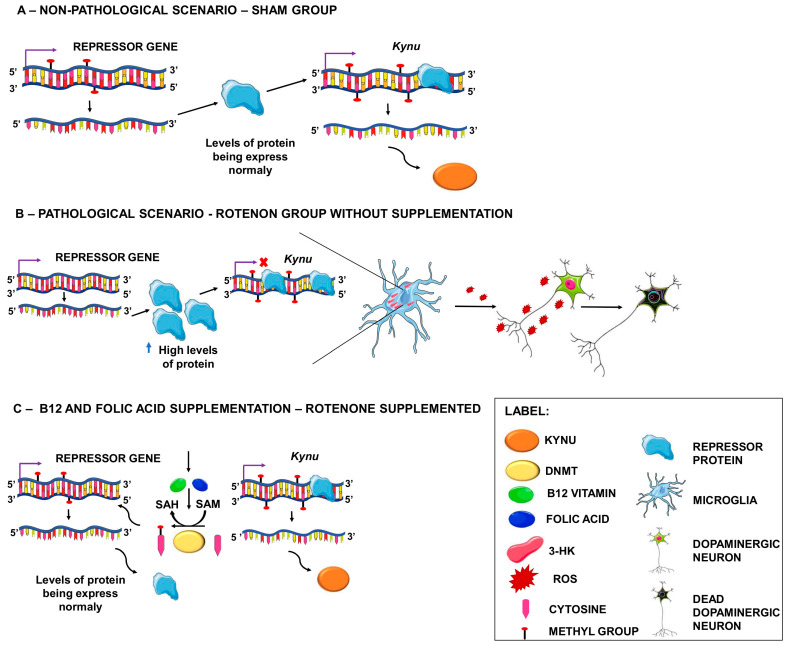
Effects of folic acid and vitamin B12 supplementation on *Kynureninase* expression. (**A**) In a non-pathological scenario, *Kynu* is possibly expressed at normal levels, as well as the repressor protein, which does not have sufficient levels to completely suppress Kynu expression. (**B**) In a pathological scenario, this repressor may have a hypomethylated promoter region, leading to continuous expression and increased protein levels, resulting in intense *Kynu* repression. KYNU can process 3-hydroxykynurenine (3-HK), and low levels of this enzyme may trigger the accumulation of 3-HK, resulting in a toxic effect due to the generation of free radical superoxide and hydrogen peroxide, which leads to cell death. (**C**) With folic acid and B12 supplementation, the DNA methylation process is normalized, and the repressor’s promoter region returns to its standard methylation state or becomes completely methylated, reducing or silencing the expression of this repressor and allowing for *Kynu* expression to occur and 3-HK to be appropriately processed.

**Figure 5 genes-13-02397-f005:**
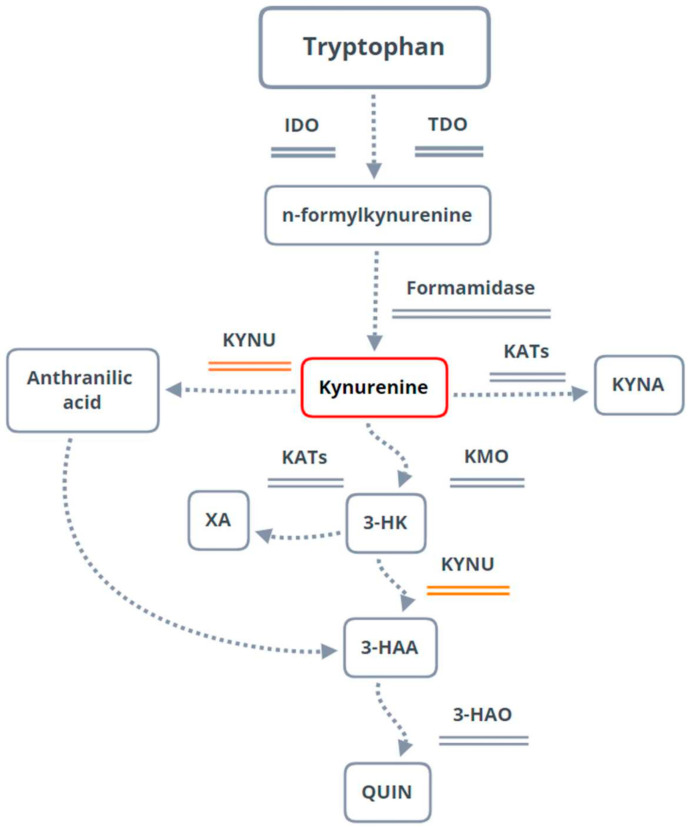
The tryptophan pathway, with emphasis on the kynurenine pathway (KP) and kynureninase (KYNU) participation. The KP is responsible for catabolizing 95% of the tryptophan available in the body [42,43]. In the brain, tryptophan is catabolized into formylkynurenine by the action of indoleamine 3-dioxygenase (IDO) and later in kynurenine (KYN) by formamidase. KYN can be processed by three different enzymes, leading to other routes. If processed by kynureninase (KYNU), the formation of anthranilic acid will occur, which can later serve as a precursor for the formation of 3-hydroxyanthranilic acid (3-HAA). If KYN is catabolized by kynurenine aminotransferases (KATs), kynurenic acid (KYNA) will form. Finally, KYN can be processed by kynurenine 3-monooxygenase (KMO), resulting in 3-hydroxykynurenine (3-HK), which can be metabolized by KYNU, giving rise to 3-HAA. XA—xanthurenic acid; 3-HA—3-hydroxyanthranilic acid 3,4-dioxygenase; QUIN—acid quinolinic; TDO—tryptophan 2,3-dioxygenase.

## Data Availability

The data presented in this study are openly available in the OSF repository at https://osf.io/4pc7u/?view_only=9061551a969047f1988bfe895e6edb85.

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
