# Peer review of "Folic Acid and Vitamin B12 Prevent Deleterious Effects of Rotenone on Object Novelty Recognition Memory and Kynu Expression in an Animal Model of Parkinson’s Disease"

_genes, 2022, doi:10.3390/genes13122397_

Round 1

Reviewer 1 Report

1. Please discuss the relevance and significance of the rodent model in Introduction

2. Animal grouping is missing from the text; even though mentioned in the Figure 1. The authors can considering re-sizing the font in Figure 1 and Figure 5.

3. Since a bilateral ICV injection was used, please provide the medial-lateral coordinates for both the sides instead of one; even if both are equidistant. This would facilitate the new researchers using this as reference in future.

4.Reference/justification for doses of folic acid/vitb12 is missing.

5.Estimation of released inflammatory markers and oxidative stress markers would have increased the strength of the study.

6. Discussion requires some more correlation between the pathophysiological effects of the model, treatments and observed results.

7. Why was a memory task used for PD instead of a motor task? Although memory impairment is a component of PD, but not as big as motor dysfunctions.

Author Response

  1. Please discuss the relevance and significance of the rodent model in Introduction

A: As suggested by the reviewer, we discussed the use of rotenone as an animal model of PD, also offering a profuse support from the literature to endorse such use.

  1. Animal grouping is missing from the text; even though mentioned in Figure 1. The authors can considering re-sizing the font in Figure 1 and Figure 5.

A: Thanks for the observation. We increased the font size of the figures to make them easier to see. Animal grouping can be found between lines 116 -121.

  1. Since a bilateral ICV injection was used, please provide the medial-lateral coordinates for both sides instead of one; even if both are equidistant. This would facilitate the new researchers using this as a reference in the future.

A: We thank the reviewer for the observation. This information is now presented in the corrected version of the manuscript.

     4. Reference/justification for doses of folic acid/vitb12 is missing.

A: In previous reports with experimental animals, it was determined that doses up to 0.1 mg/kg of vit B12 were ineffective to promote significant changes within the central nervous system (Tamaddonfard et al., 2013). In addition, another study, aiming to investigate a possible epilepsy treatment, showed that 5 mg/kg dosage of folic acid, for 7 days, was capable to restore Na+/K+-ATPase and Mg2+-ATPase activities in several brain areas of rats, also counteracting the homocysteine-thiolactone-induced seizures (Rasic-Markovic et al., 2015). Hence, this body of evidence was essential to provide elements for the doses used in our study. That is, 5 mg/Kg of folic acid and 0.5 mg/Kg of vit B12, thus, ensuring the occurrence of a significant central nervous system effect associated with the occurrence of purportedly effects on gene expressions, particularly in the Kynu gene. This justification, as requested by the reviewer, is now presented in the Material and Methods section, item 2.4. Supplementation Procedure   

      5. Estimation of released inflammatory markers and oxidative stress markers would have increased the strength of the study.

A: Indeed, this information would be valuable to corroborate our results. However, unfortunately, due to financial limitations, it was not possible to carry out such experiments.

  1. Discussion requires some more correlation between the pathophysiological effects of the model, treatments, and observed results.
  2. Why was a memory task used for PD instead of a motor task? Although memory impairment is a component of PD, but not as big as motor dysfunction.

A: Observations 6 and 7 made by the reviewer are relevant. Growing evidence shows an increase in the importance of cognitive impairment in PD. This impairment is also due to increasing life expectancy and directly impacts the quality of life of affected individuals (Fang et al., 2020, doi.org/10.1155/2020/2076942; Backstrom et al., 2018, doi.org/10.1212/WNL.0000000000006576). Moreover, cognitive decline may be considered a relevant marker of disease evolution and, together with other nonmotor disturbances, turn PD much more complex than merely a motor disease (Hobson et al., 2010, doi.org/10.1136/jnnp.2009.198689). Additionally, the rotenone-induced model of PD is typically an early-phase model, which means that its most important feature is to produce a mild set of nigrostriatal lesions. As a result, we have a model that is excellent to recapitulate nonmotor disturbances and memory dysfunctions of PD. We must say that about 90% of the literature dedicated to PD is now focusing on investigating early-phase disturbances, which makes our study perfectly aligned with the field. We added part of this discussion to the second paragraph of the discussion.

Reviewer 2 Report

I read with big interest the manuscript titled “Folic Acid and Vitamin B12 Prevent Deleterious Effects on Memory and Kynu Expression in an Animal Model of Parkinson’s Disease”. The idea of targeting kynurenine pathway as a potential therapeutic strategy in Parkinson disease has generated considerable interest in the field. The results from the current studies provide further support of the idea in the rotenone animal model. Here are my concerns:

1.      The title. I suggest the title be changed to “Folic Acid and Vitamin B12 Prevent Deleterious Effects of Rotenone on Object Novelty Recognition Memory and Kynu Expression in an Animal Model of Parkinson’s Disease”.

2.      Citations: I recommend inclusion of the following review article.

The Footprint of Kynurenine Pathway in Neurodegeneration: Janus-Faced Role in Parkinson’s Disorder and Therapeutic Implications. Tapan Behl et al. Int. J. Mol. Sci. 2021, 22,

6737. https://doi.org/10.3390/ijms22136737

3.      English: I recommend the manuscript be revised with the help of a professional English editor. I can find at least seven (7) grammar errors in the paragraph between line 75 and 89.

4.      Arrangement of control and experimental groups in Figure 2 and 3 (and supplemental figures as well). It makes more sense to start with the “Control Group”, that is S_WS; then go to R_WS, then R_Sup.

5.      Figure 2. Please consider changing the R_WS group label from “red circles” to “red triangles”. Also please label at the bottom “F(familiar)-N(new)s” into three groups: R_Sup, R_WS and S_WS.

6.      Figure 3. Please label Figure 3A with “Substantia nigra” and 3B with “Striatum”.

7.      Figure 5. Please change “3-HANA” to “3-HAA”.

8.      Methods. For the novel object recognition memory test, please cite the key reference(s).

9.      Methods. For the gene expression experiments, please describe in more detail how the gene expression level is quantified and how comparisons are made. In addition, based on the bullet point #4 above, the change should be recalculated.

Author Response

  1. The title. I suggest the title be changed to “Folic Acid and Vitamin B12 Prevent Deleterious Effects of Rotenone on Object Novelty RecognitionMemory and Kynu Expression in an Animal Model of Parkinson’s Disease”.

A: We changed the title as requested.

  1. Citations: I recommend the inclusion of the following review article.

The Footprint of Kynurenine Pathway in Neurodegeneration: Janus-Faced Role in Parkinson’s Disorder and Therapeutic Implications. Tapan Behl et al. Int. J. Mol. Sci. 2021, 22,6737. https://doi.org/10.3390/ijms22136737

A: We appreciate the recommendation and included the article in our references.

  1. 3.      English: I recommend the manuscript be revised with the help of a professional English editor. I can find at least seven (7) grammar errors in the paragraph between line 75 and 89.

A: We used the most complete Grammarly software version to solve grammatical problems.

  1. Arrangement of control and experimental groups in Figure 2 and 3 (and supplemental figures as well). It makes more sense to start with the “Control Group”, that is S_WS; then go to R_WS, then R_Sup.

A: We changed this as requested.

  1. Figure 2. Please consider changing the R_WS group label from “red circles” to “red triangles”. Also please label at the bottom “F(familiar)-N(new)s” into three groups: R_Sup, R_WS and S_WS.

A: We changed this as requested.

  1. Figure 3. Please label Figure 3A with “Substantia nigra” and 3B with “Striatum”.

A: We changed this as requested.

  1. Figure 5. Please change “3-HANA” to “3-HAA”.

A: Thanks the for the observation. We changed the figure.

  1. Methods. For the novel object recognition memory test, please cite the key reference(s).

 A: We thank the reviewer for the observation. The requested citations were properly included in the Material and Methods section, item 2.5. Object Recognition Task Test (ORT).

  1. Methods. For the gene expression experiments, please describe in more detail how the gene expression level is quantified and how comparisons are made. In addition, based on the bullet point #4 above, the change should be recalculated.

A: As requested by the reviewer, we complemented the text with more information about the gene expression comparisons (lines 214-216). All the quantification of gene expression was described between lines 206-214. We have corrected the graphics as requested.

Round 2

Reviewer 2 Report

Thanks for the revisions. I am pleased that the quality of the manuscript has been improved significantly. I agree that the article can now be published.